# Frequency and determinants of misuse of augmentation of labor in France: A population-based study

**Aude Girault**[1,2]☯*, **Béatrice Blondel**[1]☯, **François Goffinet**[1,2]☯, **Camille Le Ray**[1,2]☯

**1** INSERM UMR 1153, Obstetrical, Perinatal and Pediatric Epidemiology Research Team (Epopé), Center for Epidemiology and Statistics, FHU PREMA, Université de Paris, Paris, France, **2** Maternité Port Royal, AP-HP, Hôpital Cochin, FHU PREMA, Paris, France

☯ These authors contributed equally to this work.
* aude.girault@aphp.fr

**Data Availability Statement:** All relevant data are within the manuscript and its Supporting Information files.

## Abstract

### Introduction

While use of augmentation of labor (AL) is appropriate for labor dystocia, it is frequently used inadequately and unnecessarily. The objective was to assess at a national level, the frequency and determinants of misuse of augmentation of labor (AL).

### Material and methods

Women of the French perinatal survey of 2016 with a singleton cephalic fetus, delivering at term after a spontaneous labor were included. "Misuse of AL" was defined by artificial rupture of the membranes (ROM) and/or oxytocin within one hour of admission and/or duration between ROM and oxytocin of less than one hour. Women, labor and maternity unit's characteristics were compared between the "misuse of AL" and "no misuse of AL" groups by bivariate analysis. To identify the determinants of misuse of AL, a multivariable multilevel logistic regression was performed taking into account the data's hierarchical structure (first level: women, second level: maternity units).

### Results

Among the 7196 women included, 1524 (21.2%) had a misuse of AL. The determinants of misuse of AL were middle school educational level (reference high school), aOR = 1.21; 95%CI[1.01–1.45], gestational age at delivery ≥41weeks (reference 39–40 weeks), aOR = 1.19; 95%CI[1.00–1.42], cervical dilation ≥6cm at admission (reference <3cm), aOR = 1.39; 95%CI[1.10–1.76], epidural analgesia aOR = 1.63; 95%CI[1.35–1.96], delivery in a private hospital (reference public teaching hospital), aOR = 2.25; 95%CI[1.57–3.23]; and maternity units with <1000 deliveries/year and 1000–1999 deliveries/year (reference ≥3000 deliveries/year), respectively aOR = 1.52; 95%CI[1.11–2.08] and aOR = 1.42; 95%CI[1.05–1.92]. Less than 3% of the variance was explained by women characteristics, and 24.17% by the maternity units' characteristics.

**Funding:** This specific analysis was not funded but the collection of the analyzed data (National Perinatal Survey) was. The National Perinatal Survey was supported by the French ministry of health [Direction de la Recherche, des Études de l'Évaluation et des Statistiques (DREES), Direction Générale de la Sante (DGS) and Direction Générale de l'Organisation des Soins (DGOS)], and by Sante publique France.

**Competing interests:** The authors have declared that no competing interests exist.

## Conclusions

In France, one spontaneous laboring woman among five is subject to misuse of AL. The misuse is mostly explained by maternity unit's characteristics. The determinants identified in this study can be used to implement targeted actions in small and private maternity units.

## Introduction

Augmentation of labor (AL) using artificial rupture of the membranes (artificial ROM) and/or oxytocin infusion has been used widely since the 1960's [1, 2]. The pioneer of augmentation of labor was O'Driscoll, who described a protocol, active management of labor, aimed at achieving vaginal delivery within 12 hours of admission for all nulliparous women. This protocol included: (i) precise diagnosis of onset of labor and (ii) mandatory intervention: membrane rupture followed after one hour by oxytocin infusion, unless cervical dilatation exceeded 1 cm/hour [2]. The protocol was shown to be effective with only 4.5% women delivering after 12 hours. Since this publication, active management of labor or its components used separately have been widely studied, and have confirmed their benefit in reducing duration of labor [3–7].

Because AL does not reduce the rate of cesarean delivery [2, 3, 6, 8, 9] and could be associated with adverse maternal and neonatal outcomes such as postpartum hemorrhage, tachysystole, abnormal fetal heart rate and asphyxia [10–12], several guidelines restrict its use to labor dystocia and do not recommend it in prevention of prolonged labor. These guidelines include those from the American College of Obstetricians and Gynecologists in 2014 (ACOG), the World Health Organization in 2014 (WHO), the National Institute for Health and Care Excellence (NICE) in 2014 [13–15]. In France before 2017, no specific guidelines on use of augmentation of labor were published. Moreover, nowadays, many women have emphasized their preference towards minimal medical intervention during labor [16–18]. Restricting the use of augmentation of labor could increase maternal satisfaction regarding childbirth experience.

Previous studies have shown that AL is frequently performed inadequately or too early [12, 19, 20]. In order to restrict the use of AL to labor dystocia, it is important to identify the determinants of its misuse and implement targeted actions. These determinants could be individual such as women's characteristics, or organizational such as maternity center characteristics.

Thus, the aim of this study was to assess the frequency and determinants of misuse of augmentation of labor, using a national survey conducted in all maternity units in France.

## Methods

The studied population is from the French national perinatal survey of 2016. The French perinatal surveys are population-based studies conducted routinely every six or seven years to monitor the main indicators of perinatal health, medical practices, and risk factors. Every survey follows the same protocol, which has been described elsewhere [21]. Briefly, the sample includes all live births and stillbirths at a gestational age of at least 22 weeks or a birth weight of at least 500 g during a full week in March in all French maternity units. The design includes almost all births as less than 0.5% of births occur out of hospital [22]. Data on delivery and infant characteristics are collected from the medical records, and mothers are interviewed before their discharge to obtain maternal social and demographic characteristics and additional information about the pregnancy and their care. Each maternity unit also completes a questionnaire to provide information about its characteristics and organization.

The 2016 French National Perinatal Survey was approved by the National Council on Statistical Information (Comité du Label, 2016X703SA), the French Data Protection Authority (CNIL, 915197) and the Inserm ethics committee (IRB00003888 no. 14–191).

This analysis includes women with singleton pregnancies, who gave birth after a spontaneous labor to a live-born fetus in cephalic presentation at or after 37 weeks in mainland France. Women with a planned cesarean delivery were excluded.

We considered augmentation of labor (AL) as oxytocin infusion during labor, use of artificial rupture of membranes, or both interventions combined, in spontaneous laboring women. For the analysis, we defined "misuse of AL" by an artificial ROM within one hour of admission in the labor ward and/or an oxytocin infusion within one hour of admission and/or a duration between rupture of the membranes (ROM) and oxytocin infusion of less than one hour. Women with "no misuse of AL" were women with no artificial ROM or oxytocin augmentation during labor and women with "standard use of AL". Standard use of AL was defined by an artificial ROM at least one hour after admission in the labor ward for women with intact membranes at admission, or by an oxytocin infusion at least one hour after admission and/or by a duration between ROM and oxytocin infusion of at least one hour for women with artificial ROM or spontaneous ROM during labor. Definitions of standard use of AL and misuse of AL were constructed using previous published definitions [2, 23–27].

We first compared women's characteristics (maternal age, maternal body mass index (BMI), parity and history of cesarean delivery, country of birth, educational level, type of insurance), labor characteristics (gestational age at delivery, cervical dilation at admission, epidural analgesia) and maternity units characteristics (status, volume (number of deliveries/ year) and availability of a room dedicated to physiologic birth i.e. a room with availability of non-pharmacological methods for labor pain management such as a bathtub [28] (which was a proxy for the desire of the maternity unit to promote less medicalized births)).

In France, the law requires that maternity units must handle at least 300 deliveries a year, and there are regulations concerning the type and number of in-house staffs depending on the volume of deliveries per year. Finally, departments run by midwives are not authorized, but midwives are allowed to prescribe and administrate oxytocin and can perform artificial ROM with no medical notice. This is usually the case in public hospitals, where oxytocin and artificial ROM are usually prescribed and administrated by the midwife, without medical notice. But, in private hospitals, the physician-patient relationship leads to more decisions being made by the obstetricians, including the indication of oxytocin and artificial ROM.

To assess the determinants of misuse of augmentation of labor we performed a multivariable multilevel logistic regression taking into account the data's hierarchical structure. The characteristics of women were considered as first level and the maternity unit characteristics as second level. Variables included in the multivariable multilevel regression analysis were those known to be associated with augmentation of labor in literature and those with a p<0.20 in the bivariate analysis.

The bivariate analyses were performed with Pearson's $\chi^2$ test or Fisher's exact test when appropriate for nominal variables, and Student's t test for continuous variables. For the multivariable multilevel analysis, we used a logistic regression and tested several second-level random intercept models, adding homogeneous groups of variables. The model 0 (empty model) provided the baseline second-level variance $\tau_{00}$ assessing the variations between the maternity units of the rate of AL misuse. We then constructed two models, model 1 included women's characteristics and labor characteristics and model 2 included women's characteristics, labor characteristics and the level 2 characteristics: status and volume of the maternity units, and availability of a room dedicated to physiologic birth.

The proportional change of variance (PCV) was used to evaluate the proportion of inter-maternity unit variability $\tau_{00}$ that could be accounted for using the variables included in the models. The $\tau_{00}$ value for the models was compared to that of the previous model $(\tau_{00}(n\text{-}1) - \tau_{00}(n))/\tau_{00}(n\text{-}1)$. Adjusted Odd Ratios, aORs and 95% confidence intervals (CI), were estimated for each factor.

Data for 309 women (4.0%) were not included in the analyses because of missing data on the dependent variable i.e. oxytocin use, mode of rupture of membrane and/or timing of these interventions. The included population was comparable to the excluded population for all individual and maternity unit's characteristics.

All statistical analyses were performed with Stata (*StataCorp. 2017. Stata Statistical Software*: *Release 15. College Station*, *TX*: *StataCorp LLC*).

## Results

Among the 7 196 women included in our study, 1 524 (21.2%) had a misuse of AL during labor (Fig 1). In the misuse of AL group, 591 women (40.4%) had an artificial ROM within one hour of admission in the labor ward if the membranes were intact at admission (n = 1462), 410 (26.9%) an oxytocin infusion within one hour of admission and 857 (56.2%) a duration between ROM and oxytocin of less than an hour (S1 Table).

In the bivariate analysis, compared to women with no misuse of AL, women with a misuse of AL had higher BMIs, were less frequently multiparous with a previous cesarean delivery, had lower education level, had more frequently a cervix dilated between 3 and 5 cm at admission, had more frequently an epidural analgesia (Table 1). Women with a misuse of AL delivered more frequently in private hospitals, maternity units with <2000 deliveries/year and in units without a room dedicated to physiologic birth.

Table 2 reports the results of the multivariable multilevel logistic regression models comparing women with misuse of AL to women with no misuse of AL. The determinants associated with an increased risk of misuse of AL compared to a no misuse of AL in the complete model were middle school educational level (reference high school), aOR 1.21; 95%CI[1.01–1.45], gestational age at delivery ≥41 weeks (reference [39–40] weeks), aOR 1.19; 95%CI[1.00–

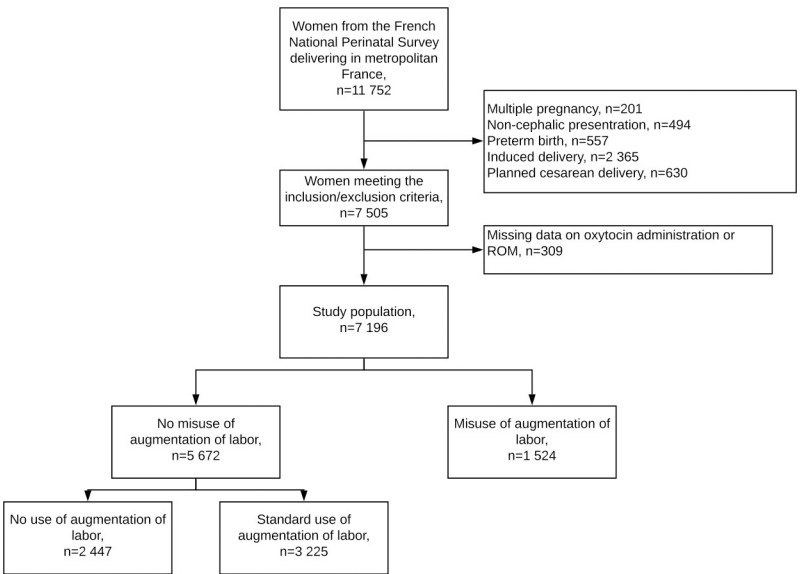

**Fig 1. Flow chart.**

**Table 1. Comparison of women's characteristics, labor characteristics and maternity unit's characteristics between women with no misuse of Augmentation of Labor (AL) and women with misuse of augmentation of labor.**

| | No misuse of AL | Misuse of AL | p |
|---|---|---|---|
| | N = 5 672 | N = 1 524 | |
| | n (%) | n (%) | |
| Maternal age, mean ± SD | 30.0 ± 4.9 | 29.8 ±5.0 | 0.17 |
| 18–25 years | 1 045 (18.4) | 302 (19.8) | 0.46 |
| 26–35 years | 3 845 (67.8) | 1 018 (66.8) | |
| >35 years | 782 (13.8) | 204 (13.4) | |
| Maternal BMI, mean ± SD | 23.3 ± 4.4 | 23.7 ± 4.6 | <**0.01** |
| <25 kg/m$^2$ | 4 101 (73.2) | 1 039 (69.0) | <**0.01** |
| [25–30] kg/m$^2$ | 1 007 (18.0) | 311 (20.7) | |
| ≥30 kg/m$^2$ | 495 (8.8) | 155 (10.3) | |
| Parity | | | **0.04** |
| Nulliparous | 2 365 (41.7) | 645 (42.4) | |
| Multiparous with no previous cesarean | 2 871 (50.6) | 791 (51.8) | |
| Multiparous with a previous cesarean | 436 (7.7) | 88 (5.8) | |
| Country of birth | | | 0.79 |
| France | 4 682 (82.6) | 1 242 (81.5) | |
| Europe | 231 (4.1) | 72 (4.7) | |
| North Africa | 369 (6.5) | 101 (6.6) | |
| Sub- Saharan Africa | 235 (4.1) | 68 (4.5) | |
| Other country | 154 (2.7) | 41 (2.7) | |
| Education level | | | **0.03** |
| Middle school | 1 186 (21.1) | 362 (24.0) | |
| High school | 1 186 (21.1) | 332 (22.0) | |
| 1 to 4 years post-graduation | 2 182 (38.7) | 539 (35.8) | |
| >4 years post-graduation | 1 075 (19.1) | 273 (18.2) | |
| Type of insurance | | | 0.66 |
| French social security | 4 931 (87.0) | 1 310 (86.0) | |
| Universal Health Insurance coverage | 608 (10.7) | 178 (11.7) | |
| State Medical Assistance | 53 (0.9) | 17 (1.1) | |
| Lack of social security coverage | 74 (1.3) | 19 (1.2) | |
| Gestational age at delivery, mean ± SD | 39.4 ±1.1 | 39.4 ±1.1 | 0.16 |
| 37–38 weeks | 1 150 (20.3) | 285 (18.7) | 0.10 |
| 39–40 weeks | 3 674 (64.8) | 981 (64.4) | |
| ≥41 weeks | 848 (14.9) | 258 (16.9) | |
| Cervical dilation on admission, cm, mean ± SD | 4.0 ± 1.9 | 4.2 ± 2.0 | <**0.01** |
| < 3 cm | 870 (15.4) | 233 (15.3) | **0.01** |
| 3–5 cm | 3 896 (69.0) | 1 087 (71.5) | |
| ≥ 6 cm | 881 (15.6) | 201 (13.2) | |
| Epidural analgesia | 4 518 (79.7) | 1 270 (83.3) | <**0.01** |
| Maternity unit status | | | <**0.01** |
| Public teaching hospital | 1 097 (19.3) | 175 (11.5) | |
| Other public hospital | 3 405 (60.0) | 855 (56.1) | |
| Private | 1 170 (20.6) | 494 (32.4) | |
| Maternity unit volume (deliveries/year) | | | <**0.01** |
| <1000 | 990 (17.5) | 334 (21.8) | |
| 1000–1999 | 1 716 (30.2) | 551 (36.2) | |
| 2000–2999 | 1 284 (22.6) | 321 (21.0) | |
| ≥3000 | 1 682 (29.7) | 318 (20.9) | |
| Maternity unit with a room dedicated to physiologic birth | 2 585 (45.6) | 643 (42.2) | **0.02** |

SD: standard deviation.

**Table 2. Association of women's socio-demographic characteristics, labor characteristics, maternity unit status and volume and misuse of augmentation of labor, multilevel model, reference: No misuse of augmentation of labor.**

| Multilevel models | Model 1* | | Model 2* | |
|---|---|---|---|---|
| | aOR | 95%CI | aOR | 95%CI |
| Level 1: women | | | | |
| Maternal age | | | | |
| 18–25 years | 1.04 | [0.88–1.24] | 1.05 | [0.88–1.25] |
| 26–35 years | Ref | - | Ref | - |
| >35 years | 1.02 | [0.85–1.23] | 1.02 | [0.85–1.23] |
| Maternal BMI | | | | |
| <25 kg/m$^2$ | Ref | - | Ref | - |
| 25–29 kg/m$^2$ | 1.15 | [0.98–1.34] | 1.15 | [0.98–1.34] |
| ≥30 kg/m$^2$ | 1.17 | [0.94–1.45] | 1.16 | [0.94–1.44] |
| Parity | | | | |
| Nulliparous | Ref | - | Ref | - |
| Multiparous with no previous of cesarean | 1.02 | [0.89–1.17] | 1.02 | [0.89–1.18] |
| Multiparous with history of cesarean | **0.72** | **[0.55–0.93]** | **0.72** | **[0.55–0.94]** |
| Education level | | | | |
| Middle school | **1.21** | **[1.01–1.44]** | **1.21** | **[1.01–1.45]** |
| High school | Ref | - | Ref | - |
| 1 to 4 years post-graduation | 1.10 | [0.92–1.30] | 1.10 | [0.92–1.30] |
| >4 years post-graduation | 1.06 | [0.89–1.27] | 1.08 | [0.90–1.29] |
| Type of health security | | | | |
| French social security | Ref. | - | Ref. | - |
| Universal Health Insurance coverage | 1.07 | [0.87–1.33] | 1.10 | [0.89–1.36] |
| State Medical Assistance | 1.37 | [0.74–2.55] | 1.49 | [0.80–2.77] |
| Lack of social security coverage | 1.03 | [0.58–1.83] | 1.10 | [0.62–1.96] |
| Gestational age at delivery, | | | | |
| 37–38 weeks | 0.92 | [0.79–1.09] | 0.91 | [0.78–1.07] |
| 39–40 weeks | Ref | - | Ref | - |
| ≥41 weeks | 1.18 | [0.99–1.40] | **1.19** | **[1.00–1.42]** |
| Cervical dilatation at admission, | | | | |
| <3 cm | Ref | - | Ref | - |
| 3–5 cm | 0.88 | [0.74–1.06] | 0.90 | [0.75–1.08] |
| ≥6 cm | **1.37** | **[1.08–1.73]** | **1.39** | **[1.10–1.76]** |
| Epidural analgesia | **1.61** | **[1.33–1.94]** | **1.63** | **[1.35–1.96]** |
| Level 2: maternity units | | | | |
| Maternity unit status | | | | |
| Public teaching hospital | | | Ref | - |
| Public hospital | | | 1.36 | [0.97–1.89] |
| Private | | | **2.25** | **[1.57–3.23]** |
| Maternity unit volume (deliveries/year) | | | | |
| <1000 | | | **1.52** | **[1.11–2.08]** |
| 1000–1999 | | | **1.42** | **[1.05–1.92]** |
| 2000–2999 | | | 1.15 | [0.84–1.58] |
| ≥3000 | | | Ref | - |
| Maternity unit with a room dedicated to physiologic birth | | | 0.85 | [0.70–1.02] |
| PCV† (%) | | 2.69 | | 24.17 |

*Model 1 includes women's characteristics and labor characteristics.

Model 2 includes women's characteristics, labor characteristics and the level 2 characteristics: status and volume (number of deliveries/year) of the maternity units, and maternity unit with a room dedicated to physiologic birth.

†PCV: proportional change of variance (PCV), use to evaluate the proportion of inter-maternity unit variability that can be accounted for using the variable of the models.

1.42], cervical dilatation ≥6cm at admission (reference cervix dilated <3cm), aOR 1.39; 95% CI[1.10–1.76], epidural analgesia aOR 1.63; 95%CI[1.35–1.96], delivery in a private hospital (reference public teaching hospital), aOR 2.25; 95%CI[1.57–3.23]; and maternity units with <1000 deliveries/year and [1000–2000] deliveries/year (reference ≥3000 deliveries/year), aOR 1.52; 95%CI[1.11–2.08] and aOR 1.42; 95%CI[1.05–1.92] respectively. One determinant, multiparous women with a previous cesarean delivery was associated with a lower probability of having a misuse of AL, aOR 0.72; 95%CI[0.55–0.94]. Less than 3% of the variance was explained by the first model i.e. the model including maternal and labor characteristics. The complete model showed that 24.17% of the variance was explained by the maternity units' characteristics.

## Discussion

### Main findings

This study shows that misuse of augmentation of labor is frequent and has specific maternal determinants: admission in labor ward during the active phase of labor (i.e. after a cervical dilation of 5cm), epidural analgesia and gestational age ≥ 41 weeks. However, misuse is mostly explained by the maternity unit's characteristics. It is more frequent among women delivering in private hospitals and in maternity units with <2000 deliveries/year.

### Strengths and limitations

It is to our knowledge the first study aimed at identifying determinants of misuse of AL. The French Perinatal Survey is a population-based study with a low rate of missing data and good quality data as they were collected by technician research midwives. As the survey includes all maternity units in France, our results cover the diversity of medical practices in this country and the overall sample is representative of all annual births in France [21]. The number of determinants studied i.e. individual characteristics, labor characteristics and maternity unit's characteristics allow identifying determinants of misuse of AL and thus, subgroups of women in which targeted actions could be implemented to decrease misuse of AL.

The main limitation of this study is the lack of information on indication of use of AL. Indeed, the purpose of the French national perinatal surveys is to provide data on a wide range of topics related to perinatal health, risk factors, medical practices and preventive behavior; consequently, it was not planned to collect detailed data on indication of AL. This lack of information prevents us from further investigating the indications of AL, and could have led to a classification bias. For example, in a woman admitted at 8 cm with abnormal fetal heart rate, artificial ROM or oxytocin could indeed be indicated to shorten labor as soon as the women enter the labor ward, it is therefore not a misuse of AL. But, a woman receiving augmentation of labor two hours after entering the labor ward with a 3cm cervical dilation was not considered as having a misuse of AL. Thus, we were not able to specifically identify such situations. Our definition of misuse of AL includes both mis-indicated use and mis-dispensation of AL and tends to underestimate the rate of misuse of augmentation of labor without affecting the interpretation of the observed association.

In addition, even though the adopted definition of misuse of AL has been previously utilized in published studies, it can be discussed as the definition of labor dystocia has evolved with time and the use of AL can today be delayed. Indeed, in O'Driscoll's active management of labor "precise diagnosis of onset of labor" was a crucial point to decide on the "mandatory interventions" if the cervix did not dilate at 1cm/hour. In our sample we have no information on what happened before admission in labor ward; it is possible that some women were in labor before entering the labor ward.

To this day there is no clear definition of labor dystocia all the more in the latent first stage [14] and in France the guidelines on when to start augmentation of labor were issued after the present study (2017) [23]. In any case, the definition used in our study (one hour after admission to start AL and/or one hour between oxytocin and rupture of membranes) is restrictive and could underestimate the rate of misuse of AL; but it lowers the risk of including standard use of AL in the group of misuse.

## Interpretation

Obstetric characteristics associated with misuse of AL may reflect the will of physicians to reduce labor duration of women known to have longer labors: women delivering ≥41weeks and women with an epidural analgesia. Nevertheless, the association of misuse of AL and advanced cervical dilatation (i.e. women with a cervical dilation ≥ 6cm at admission) is in conflict with that hypothesis. Even though there is no medical justification to AL use, limiting pain duration by shortening labor among these women is a possible explanation for this association. Unfortunately, the rate of women reporting a written birth project in the French perinatal survey of 2016 was low (4.2%) with no differences between the two groups, and the details of the project (i.e. desire for low interventional birth) were not reported in the survey.

The hypothesis to explain the association between low educational level and misuse of AL could be that low educated women are less frequently in control of their care and are less in demand of a birth without medical interventions [16]. However, in the end, individual characteristics only explain a small part of the inter-maternity variability.

Finally, maternity unit's characteristics, which reflect the units' organization and policies, were the main identified determinants of misuse of labor in our study. We indeed observed an association between misuse of AL and both status and volume of the maternity units. This finding is consistent with those of a previous French study which included low obstetric risk women and showed that the use of oxytocin was associated with the same two factors [10]. Constraints related to the practice in private hospitals could partly explain the increase of misuse of AL in these hospitals. Indeed, in many private French hospitals, obstetricians attend both the births of their patients and private consultations for other patients sometimes outside the hospital. Because of these constraints, as it has been suggested for the increase in operative vaginal deliveries in theses settings, we hypothesize that augmentation of labor can facilitate their time-management [29, 30].

Maternity units with low to moderate volume of deliveries are also confronted with the availability of the medical team (anesthetist, obstetrician, and pediatrician) because their presence is not permanent. Another explanation could come from the greater degree of adherence to evidence-based medicine in high volume units and in the public hospitals. This has already been described for other medical practices such as tocolysis and postpartum hemorrhage prevention [10, 31, 32]. The high-volume maternity units are more frequently university hospitals and are particularly attentive to following guidelines. Furthermore, in France, guidelines on augmentation of labor were published in 2017 by the midwives' college and the obstetricians and gynecologists' college, these guidelines were mostly drafted by health staff working in public university hospitals [23]. In addition, it is known that midwives, who have great autonomy in the management of labor in public maternity units, are less favorable to augmentation of labor [33, 34].

Another result supports the importance of the policies followed by the units. The maternity centers with a unit supporting physiologic birth have indeed less misuse of AL than other centers. This unit's characteristic shows the willingness of the maternity center to promote less medicalized childbirth or at least a more adequate medicalized childbirth.

As reducing labor duration could be required during peak periods of activity, the workload in the labor ward during the survey could be informative on why early AL would be performed. This determinant could not be studied, as only information on status of maternity unit and maternity volume were available to study. One hypothesis could be that the maternity units most inclined at performing misuse of AL would be those trying to speed up labor in order to free up beds. It has been shown in a Swedish study, that pressure from other midwives or obstetricians, and shortage of delivery rooms are factors influencing the decision of starting AL [35].

Misuse of augmentation of labor is not insignificant as it is known that AL can be associated with maternal and fetal consequences such as postpartum hemorrhage, tachysystole, abnormal fetal heart rate and asphyxia [11, 12, 36]. In addition, in the context of women's increasing desire for natural childbirth, and guidelines promoting reduction of unnecessary medical intervention [15, 37, 38], it is important to inform physicians working in small and private maternity units and to implement targeted actions to reduce misuse of AL rates[16–18].

## Conclusion

This study showed that misuse of AL occurred in nearly one third of spontaneous laboring women receiving augmentation of labor in France. The misuse seemed to be mostly explained by maternity unit's characteristics. The identification of the determinants associated with misuse of AL allows us to specifically target maternity units to whom the recently published guidelines apply, i.e. small and private maternity units and maternity center without a unit supporting physiologic birth, in order to offer them suitable training.

## Supporting information

**S1 Table. Rate and characteristics of rupture of the membranes and oxytocin infusion in the groups of augmentation of labor.** AL = Augmentation of Labor; NA = not applicable. (PDF)

## Acknowledgments

The authors thank the Maternal and Child Health services in each district, the department heads in each maternity unit and the investigators who allowed data collection, and the women who agreed to be interviewed.

## Author Contributions

**Conceptualization:** Aude Girault, Camille Le Ray.

**Formal analysis:** Aude Girault, Camille Le Ray.

**Methodology:** Aude Girault, Béatrice Blondel, François Goffinet, Camille Le Ray.

**Software:** Aude Girault.

**Supervision:** François Goffinet, Camille Le Ray.

**Validation:** Aude Girault, Camille Le Ray.

**Writing – original draft:** Aude Girault, Camille Le Ray.

**Writing – review & editing:** Aude Girault, Béatrice Blondel, François Goffinet, Camille Le Ray.

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
