## [Decision Letter · Decision Letter 0]

7 Jan 2021

PONE-D-20-38267

Frequency and determinants of misuse of augmentation of labor in France: a population-based study

PLOS ONE

Dear Dr. Girault,

Thank you for submitting your manuscript to PLOS ONE. After careful consideration, we feel that it has merit but does not fully meet PLOS ONE’s publication criteria as it currently stands. Therefore, we invite you to submit a revised version of the manuscript that addresses the points raised during the review process.

We look forward to receiving your revised manuscript.

Kind regards,

David Desseauve, MD, MPH, PhD

Academic Editor

PLOS ONE

Journal Requirements:

Reviewers' comments:

Reviewer's Responses to Questions

**Comments to the Author**

1. Is the manuscript technically sound, and do the data support the conclusions?

Reviewer #1: Yes

Reviewer #2: Partly

Reviewer #3: Yes

2. Has the statistical analysis been performed appropriately and rigorously? 

Reviewer #1: Yes

Reviewer #2: Yes

Reviewer #3: Yes

3. Have the authors made all data underlying the findings in their manuscript fully available?

Reviewer #1: No

Reviewer #2: Yes

Reviewer #3: Yes

4. Is the manuscript presented in an intelligible fashion and written in standard English?

Reviewer #1: Yes

Reviewer #2: Yes

Reviewer #3: Yes

5. Review Comments to the Author

Reviewer #1: Thanks for the opportunity to review this very interesting work. The authors aimed to investigate French practices about augmentation of labor in a national French Database. They report than 1 woman in 5 receive inadequately an augmentation of labor and that most of factors associated with this misuse are organizational ones. The paper is a good quality one but some points deserve to be more detailed/discussed. The main outcome definition might be more precise. Some aspects of the discussion should be more developed and a message about clinical prospects is missing.

woman in 5 receive inadequately an augmentation of labor and that most of factors associated with this misuse are organizational ones. The paper is a good quality one but some points deserve to be more detailed/discussed. The main outcome definition might be more precise. Some aspects of the discussion should be more developed and a message about clinical prospects is missing.

You will find my detailed comments below

Detailed comments

Introduction

- By reading the introduction, the hypothesis remains unclear

- Considering that it is a French national database study, it might be interesting to report explicitly at this state of the paper what are the French guidelines about augmentation of labor

- Another point not addressed in the direction is that more and more women intend to deliver with the minimalist medical intervention which is discordant with a large use of induction of labor. It is likely that controlling the use of AL (without increasing the rate of C-section) will improve women’s satisfaction about childbirth.

Methods

- The delay of one hour after admission is confusing lines 95-96. Did you mean one hour after admission into the maternity? One hour after admission into labor ward?

- Why not considering a cut-off about cervical dilatation? Indeed, women could be admitted in labor ward even if they are not in active labor for pain management. In such a situation: a woman receiving ROM+oxytocin 2 hours after admission for stagnation at 3cm of dilatation will not be considered as “misuse of AL” whereas it is clearly one

- One limitation is that you are not able to report the indication of AL. Most of the time it is for dystocia, but sometimes AL can be used for suspect fetal heart. It is difficult to consider as “misuse of AL” the case of a women with ROM at 6cm within the first hour within her admission in case of abnormal fetal heart rate (in order to perform fetal pH for example)

- Don’t you think that your definition of AL summarizes two outcomes: misuse of AL (use of AL without indication) and wrong use of AL modalities (less than one hour between ROM and oxytocin administration)?

Results

- As explained above It is difficult to interpret your analysis about cervical dilatation. A women receiving ROM + oxytocin at 3 cm, 2 hours after her admission in labor ward at the same dilatation is considered as “adequate use of AL”?

- Considering that there is no difference between university and non university hospital, don’t you think an analysis public vs private hospital could be more informative? It allowed to keep in the analysis the most important part of your population

Discussion and conclusion

- ¬Ok for the discussion about indication of AL suggested in one of my previous comment

- Line 306: it would have been interesting to have the information of the proportion of women with a written birth project requiring a “low interventional” birth in their obstetric file.

- The interpretation about public vs private practice is difficult. I think that it is possible that difference is more associated with the professional taking the decision: MD are probably more inclined to use AL than midwives. Is there any foreign literature (especially UK literature regarding their health service structuration)?

- I think a message about the prospects is missing. You report that 1 woman in 5 receive inadequately an augmentation of labor? What are your suggestions to improve practices?

Reviewer #2: In this manuscript the authors present a retrospective cohort study that reports the misuse of labor augmentation in France. The authors included women of the 2016 French perinatal survey with a term singleton pregnancy with a spontaneous labor. They define misuse of augmentation of labor (AL) as an artificial rupture of membranes within one hour of admission, and/or an oxytocin infusion within one hour of admission and/or a duration between rupture of membranes and oxytocin infusion of less than one hour. The authors reported the percentage of misuse of AL in French maternities and the determinants of misuse of AL after a multivariable analysis.

This study comports a major classification bias as for the definition of misuse of labor, limiting the interpretation of the results. More commonly misuse of AL is define in cases where no dystocia of labor was demonstrated (Wei S et al. Cochrane Database Syst Rev. 2013 / Selin et al. Acta Obstet Gynecol Scand. 2009). In this study, it is unknown if the patient classified in misuse of AL presented with dystocia of labor. Furthermore, as stated by the authors in the discussion they could access the indication for either the introduction of Oxytocin nor amniotomy. Probably, a part of the women in the group misuse of labor had a medical necessity for intervention such as non-reassuring fetal heart rates, chorioamniotitis, pre-eclampsia, bleeding of unknown origin. It could also be viewed that the 15% women presenting with a cervix < 3 cm, as most authors currently define active labor as either a cervical dilatation > 4 or 6 cm. Those women might have had an indication for labor induction.

Secondly, this study does not investigate the maternal nor neonatal outcomes associated with misuse of AL, which could have been interesting. The authors reported the determinant of misuse of AL in France. The results of this study is beyond the scope of an international journal as it focuses solely on reporting French labor ward practices. Organization and management of the labor ward differ from one country to another, these determinants could not be translated internationally.

Reviewer #3: The aim of this national study was to was to assess the frequency and determinants of misuse of augmentation of labor. All the data are extracted from a national survey of 2016 concerning women at term with a spontaneous labor and singleton, cephalic presentation. The topic of this paper is very interesting in the context of tendency of a limitation of medical intervention during labor.

This study involves 7196 women from different French private or public maternities and provide an overview of French practices. The authors found a rate of 20% of mis-use of oxytocin. This the first French national study providing this result which can help all the maternities to improve their practices by comparing their own rate. For the authors, the misuse of AL seemed to be mostly explained by maternity unit’s characteristics, especially private hospital and maternities with less 1000 deliveries/ year.

The main limitation of the study is the definition of misuse of AL. The authors remember that there is no international and consensual definition of misuse of augmentation of labor . Thus, they propose their own definition which can be a little restrictive without distinguishing passive and active first stage of labor. The references gien by the authors to justify their definition are old ( 1990’s). Misuse is mostly define by the time interval between admission and use of oxytocin or artificial ROM ( less than 1h). Thus, it is possible that misuse of AL is underestimated in this study. The authors explained this limitation in the discussion. They noted that misuse of AL was encountered in near 13% of women admitted with a cervical dilation > 6cm. Limitation of pain duration could be an argue, but authors should not forget that this survey was conducted just before publication of French Guidelines concerning use of oxytocine. Before this publication, use of oxytocin was just only a “work habit” without established scientific evidence. A similar work would be interesting using similar data of the next French national survey.

The paper and tables are well written and easy to read.

6. PLOS authors have the option to publish the peer review history of their article (what does this mean?). If published, this will include your full peer review and any attached files.

Reviewer #1: No

Reviewer #2: No

Reviewer #3: No

---

## [Author Response · Author response to Decision Letter 0]

19 Jan 2021

January 19th, 2021

Dear Editor,

We submit for your consideration for publication in Plos One our revised manuscript entitled “Frequency and determinants of misuse of augmentation of labor in France: a population-based study” (PONE-D-20-38267). 

The authors are very grateful to the Reviewers for their constructive help. We think the paper has been much improved. 

Each point raised by the referees and editors has been answered, and the manuscript revised accordingly. Responses of the authors are included below after each comment. The position of all changes made in the manuscript is indicated with “track changes”.

All the authors have read and approved the revised version of the paper.

We hope our manuscript now meets the standards of Plos One.

Yours sincerely,

Aude Girault

https://clicktime.symantec.com/3WtNZRwBkHqvQ84YqC4gX5z6H2?u=https%3A%2F%2Fjournals.plos.org%2Fplosone%2Fs%2Ffile%3Fid%3DwjVg%2FPLOSOne_formatting_sample_main_body.pdf

https://clicktime.symantec.com/3rTEauhYA7zZceCAdkgwUt6H2?u=https%3A%2F%2Fjournals.plos.org%2Fplosone%2Fs%2Ffile%3Fid%3Dba62%2FPLOSOne_formatting_sample_title_authors_affiliations.pdf

 The authors have ensured that the manuscript met Plos One’s style requirements.

The authors have removed the phrase data not shown as these data are not a core part of the research.

Reviewer's Responses to Questions

Comments to the Author

Reviewer #1: Thanks for the opportunity to review this very interesting work. The authors aimed to investigate French practices about augmentation of labor in a national French Database. They report than 1 woman in 5 receive inadequately an augmentation of labor and that most of factors associated with this misuse are organizational ones. The paper is a good quality one but some points deserve to be more detailed/discussed. The main outcome definition might be more precise. Some aspects of the discussion should be more developed and a message about clinical prospects is missing.

woman in 5 receive inadequately an augmentation of labor and that most of factors associated with this misuse are organizational ones. The paper is a good quality one but some points deserve to be more detailed/discussed. The main outcome definition might be more precise. Some aspects of the discussion should be more developed and a message about clinical prospects is missing.

You will find my detailed comments below

Detailed comments

Introduction

- By reading the introduction, the hypothesis remains unclear

The authors thank the reviewer for his comment, this study was exploratory, therefore there were no hypotheses favoring one specific determinant over the others. The only hypothesis line 71 to 73 was that the “determinants could be individual such as women’s characteristics, or organizational such as maternity center characteristics”.

- Considering that it is a French national database study, it might be interesting to report explicitly at this state of the paper what are the French guidelines about augmentation of labor

The authors agree with the reviewer and have modified their manuscript accordingly, the modified version of the manuscript now states lines 67-68: “In France before 2017, no specific guidelines on use of augmentation of labor were published.” Nevertheless, as explained lines 64 to 67, international guidelines existed at the time of the study. 

- Another point not addressed in the direction is that more and more women intend to deliver with the minimalist medical intervention which is discordant with a large use of induction of labor. It is likely that controlling the use of AL (without increasing the rate of C-section) will improve women’s satisfaction about childbirth.

The authors agree with the reviewer, and have added a sentence in the modified version of their manuscript, line 70-71 “Restricting the use of augmentation of labor could increase maternal satisfaction regarding childbirth experience”.

Methods

- The delay of one hour after admission is confusing lines 95-96. Did you mean one hour after admission into the maternity? One hour after admission into labor ward?

The authors have clarified the sentence which now states line 100 “in the labor ward”

- Why not considering a cut-off about cervical dilatation? Indeed, women could be admitted in labor ward even if they are not in active labor for pain management. In such a situation: a woman receiving ROM+oxytocin 2 hours after admission for stagnation at 3cm of dilatation will not be considered as “misuse of AL” whereas it is clearly one

The authors understand the reviewer’s point who underlines one of the limits of their study. Indeed, it was impossible for them to differentiate the women admitted in the labor ward for pain management from the women admitted for a “real” labor onset. Even though the definition used in this article is restrictive, it is a practical definition which limits the overestimation of misuse of augmentation of labor. If we added a cut-off for cervical dilation the risk would be to overestimate the rate of misuse of AL and therefore identify ”unreal” associations. Indeed, if all women with a cervical dilation under 6 cm receiving augmentation of labor were considered as having misuse of AL, misuse of AL would concern one in two women. Moreover, the cervical dilation was taken in account in the analyses. For the reviewer’s information a stratified analysis on cervical dilation (cervix <6cm / �6 cm) was performed by the authors and found the same determinants. 

- One limitation is that you are not able to report the indication of AL. Most of the time it is for dystocia, but sometimes AL can be used for suspect fetal heart. It is difficult to consider as “misuse of AL” the case of a women with ROM at 6cm within the first hour within her admission in case of abnormal fetal heart rate (in order to perform fetal pH for example)

The authors totally agree with the reviewer and have underlined this in their discussion lines 281-291: “This lack of information prevents us from further investigating the indications of AL, and could have led to a classification bias. For example, in a woman admitted at 8 cm with abnormal fetal heart rate, artificial ROM or oxytocin could indeed be indicated to shorten labor as soon as the women enter the labor ward, it is therefore not a misuse of AL. But, a woman receiving augmentation of labor two hours after entering the labor ward with a 3cm cervical dilation was not considered as having a misuse of AL. Thus, we were not able to specifically identify such situations. Our definition of misuse of AL includes both mis-indicated use and mis-dispensation of AL and tends to underestimate the rate of misuse of augmentation of labor without affecting the interpretation of the observed association.”

- Don’t you think that your definition of AL summarizes two outcomes: misuse of AL (use of AL without indication) and wrong use of AL modalities (less than one hour between ROM and oxytocin administration)?

The authors agree with the reviewer’s comment, our definition includes both mis-indicated use and mis-dispensation of augmentation of labor, this is why the authors chose to name it misuse of augmentation of labor. In order to clarify this point, the authors have added a sentence lines 289-290 of their modified manuscript: “Our definition of misuse of AL includes both mis-indicated use and mis-dispensation of AL”

Results

- As explained above It is difficult to interpret your analysis about cervical dilatation. A women receiving ROM + oxytocin at 3 cm, 2 hours after her admission in labor ward at the same dilatation is considered as “adequate use of AL”?

Indeed, as explained above, the authors chose a restrictive definition of misuse of AL with no specification of indications. The definition chosen probably underestimated the rate of misuse of AL and the observed associations. This has been added in the modified version of the manuscript lines 286-291: “But, a woman receiving augmentation of labor two hours after entering the labor ward with a 3cm cervical dilation was not considered as having a misuse of AL. Thus, we were not able to specifically identify such situations. Our definition of misuse of AL includes both mis-indicated use and mis-dispensation of AL and tends to underestimate the rate of misuse of augmentation of labor without affecting the interpretation of the observed association.” 

- Considering that there is no difference between university and non university hospital, don’t you think an analysis public vs private hospital could be more informative? It allowed to keep in the analysis the most important part of your population

The authors thank the reviewer for his suggestion, but as they were searching for determinants of misuse of AL the use of the three-category variable for maternity unit status (university public, non-university, private) allowed to keep all of the population in the analysis. Moreover, the authors feel that analyzing the maternity unit status as a three-category variable allows to compare the association between misuse of AL and public university and public non-university hospitals. The fact that there is no difference between public university and public non-university hospitals is indeed an interesting information.

Discussion and conclusion

- ¬Ok for the discussion about indication of AL suggested in one of my previous comment

The authors thank the reviewer for his comment.

- Line 306: it would have been interesting to have the information of the proportion of women with a written birth project requiring a “low interventional” birth in their obstetric file.

The authors totally agree with the reviewer, unfortunately the women reporting a written birth project was low in the 2016 national survey (3.7%) and in our population (4.2%) with no difference between the women with no differences between the two groups. The details of the project were not reported in the French Perinatal survey, therefore it is impossible to know how many women had a project requiring a “low interventional” birth. This information was added lines 316 to 319 of the modified manuscript: “Unfortunately, the rate of women reporting a written birth project in the French perinatal survey of 2016 was low (4.2%) with no differences between the two groups, and the details of the project (i.e. desire for low interventional birth) were not reported in the survey.”

- The interpretation about public vs private practice is difficult. I think that it is possible that difference is more associated with the professional taking the decision: MD are probably more inclined to use AL than midwives. Is there any foreign literature (especially UK literature regarding their health service structuration)?

The authors agree with the reviewer’s point as they have underlined lines 345-347 of their discussion: “midwives, who have great autonomy in the management of labor in public maternity units, are less favorable to augmentation of labor [33,34].”, but to their knowledge there are no foreign literature on the subject. Moreover, in the French Perinatal Survey there were no information on who prescribed augmentation of labor.

- I think a message about the prospects is missing. You report that 1 woman in 5 receive inadequately an augmentation of labor? What are your suggestions to improve practices?

The authors thank the reviewer for his comment. They have modified their conclusion as follow lines 373 to 376: “The identification of the determinants associated with misuse of AL allows us to specifically target maternity units to whom the recently published guidelines apply, i.e. small and private maternity units and maternity center without a unit supporting physiologic birth, in order to offer them suitable training.”

Reviewer #2: In this manuscript the authors present a retrospective cohort study that reports the misuse of labor augmentation in France. The authors included women of the 2016 French perinatal survey with a term singleton pregnancy with a spontaneous labor. They define misuse of augmentation of labor (AL) as an artificial rupture of membranes within one hour of admission, and/or an oxytocin infusion within one hour of admission and/or a duration between rupture of membranes and oxytocin infusion of less than one hour. The authors reported the percentage of misuse of AL in French maternities and the determinants of misuse of AL after a multivariable analysis.

This study comports a major classification bias as for the definition of misuse of labor, limiting the interpretation of the results. More commonly misuse of AL is define in cases where no dystocia of labor was demonstrated (Wei S et al. Cochrane Database Syst Rev. 2013 / Selin et al. Acta Obstet Gynecol Scand. 2009). In this study, it is unknown if the patient classified in misuse of AL presented with dystocia of labor.

Furthermore, as stated by the authors in the discussion they could access the indication for either the introduction of Oxytocin nor amniotomy. Probably, a part of the women in the group misuse of labor had a medical necessity for intervention such as non-reassuring fetal heart rates, chorioamniotitis, pre-eclampsia, bleeding of unknown origin. 

The authors understand the reviewer’s point. Indeed, some women may have had complications during labor indicating AL. It is therefore possible that the rate of misuse of AL was overestimated but, only women at term, in spontaneous labor were included which limits this risk. Moreover, our population was of low obstetric risk with only 0.35% of preeclampsia and 1.3% gestational hypertension with no differences between the two groups. 

It could also be viewed that the 15% women presenting with a cervix < 3 cm, as most authors currently define active labor as either a cervical dilatation > 4 or 6 cm. Those women might have had an indication for labor induction.

The authors thank the reviewer for his comment. Indeed, the definition of active labor could be discussed, but again only women considered in spontaneous labor were included. Moreover, all medical files are reviewed by technician research midwives (lines 270-271), who are independent from the maternity unit which allow to affirm that the patients were considered in spontaneous labor when entering the labor ward.

Secondly, this study does not investigate the maternal nor neonatal outcomes associated with misuse of AL, which could have been interesting. 

The authors understand the reviewers point, they feel that outcomes of misuse of augmentation of labor should be the subject of a separate paper. The analyses on the outcomes have been performed and are the subject of a separate paper currently being written. In view of the frequency of misuse of augmentation of labor, the authors believe that the study of its determinants is a separate subject to be dealt with in an independent manner.

The authors reported the determinant of misuse of AL in France. The results of this study is beyond the scope of an international journal as it focuses solely on reporting French labor ward practices. Organization and management of the labor ward differ from one country to another, these determinants could not be translated internationally.

The authors disagree with the reviewer’s comment. The too frequent use and the misuse of oxytocin and amniotomy is an international issue as shown by the international literature and the recent international guidelines. There is to this day and to the authors’ knowledge no such population-based study which are the most adequate type of studies to investigate determinants. The French determinants identified in this study could be, at least partly, translated internationally (admission in labor ward during the active phase of labor, epidural analgesia and gestational age) and fully in countries with a similar maternity ward organization. 

Reviewer #3: The aim of this national study was to was to assess the frequency and determinants of misuse of augmentation of labor. All the data are extracted from a national survey of 2016 concerning women at term with a spontaneous labor and singleton, cephalic presentation. The topic of this paper is very interesting in the context of tendency of a limitation of medical intervention during labor.

This study involves 7196 women from different French private or public maternities and provide an overview of French practices. The authors found a rate of 20% of mis-use of oxytocin. This the first French national study providing this result which can help all the maternities to improve their practices by comparing their own rate. For the authors, the misuse of AL seemed to be mostly explained by maternity unit’s characteristics, especially private hospital and maternities with less 1000 deliveries/ year.

The main limitation of the study is the definition of misuse of AL. The authors remember that there is no international and consensual definition of misuse of augmentation of labor . Thus, they propose their own definition which can be a little restrictive without distinguishing passive and active first stage of labor. The references gien by the authors to justify their definition are old ( 1990’s). Misuse is mostly define by the time interval between admission and use of oxytocin or artificial ROM ( less than 1h). Thus, it is possible that misuse of AL is underestimated in this study. The authors explained this limitation in the discussion. They noted that misuse of AL was encountered in near 13% of women admitted with a cervical dilation > 6cm. Limitation of pain duration could be an argue, but authors should not forget that this survey was conducted just before publication of French Guidelines concerning use of oxytocine. Before this publication, use of oxytocin was just only a “work habit” without established scientific evidence. A similar work would be interesting using similar data of the next French national survey.

The paper and tables are well written and easy to read.

The authors thank the reviewer for his comments. Even though there were no guidelines concerning augmentation of labor in France before 2017, active management of labor was not recommended and international guidelines existed at the time of the study. A similar work using the data for the next French national survey which will be conducted in march 2021 (data available in 2022), will be carried out to evaluate the potential impact of the recent French guidelines. The present recent data provide a status report of rate of misuse of AL before the publication of the French guidelines and enable to inform immediately the obstetrical teams of the associated determinants.

---

## [Editor Report · Decision Letter 1]

26 Jan 2021

Frequency and determinants of misuse of augmentation of labor in France: a population-based study

PONE-D-20-38267R1

Dear Dr. Girault,

We’re pleased to inform you that your manuscript has been judged scientifically suitable for publication and will be formally accepted for publication once it meets all outstanding technical requirements.

Kind regards,

David Desseauve, MD, MPH, PhD

Academic Editor

PLOS ONE

---

## [Editor Report · Acceptance letter]

29 Jan 2021

PONE-D-20-38267R1 

Frequency and determinants of misuse of augmentation of labor in France: a population-based study 

Dear Dr. Girault:

I'm pleased to inform you that your manuscript has been deemed suitable for publication in PLOS ONE. Congratulations! Your manuscript is now with our production department. 

Kind regards, 

on behalf of

Dr. David Desseauve 

Academic Editor

PLOS ONE